

# A natural composite Higgs via universal boundary conditions

**Simone Blasi**[1][⋆], **Csaba Csáki**[2][†] **and Florian Goertz**[1][‡]

**1** Max-Planck-Institut für Kernphysik, Saupfercheckweg 1, 69117 Heidelberg, Germany
**2** Department of Physics, LEPP, Cornell University, Ithaca, NY 14853, USA

⋆ simone.blasi@mpi-hd.mpg.de, † csaki@cornell.edu, ‡ fgoertz@mpi-hd.mpg.de

## Abstract

We present a novel realization of a composite Higgs, which can naturally produce top partners above the current LHC bounds without increasing the tuning above 10%. The essential ingredients are softened breaking of the Higgs shift symmetry as well as maximal symmetry, which turn out to perfectly complement each other. The 5D realization of this model is particularly simple: universal UV and IR boundary conditions for the bulk fermions containing the SM fields will cure the problems of existing holographic composite Higgs models and provide a complete viable model for a naturally light Higgs without much tuning.

# 1  Introduction

The continued absence of New Physics (NP) discoveries at the LHC makes the lightness of the Higgs boson even more mysterious. One would expect the appearance of NP at the TeV scale that protects the Higgs mass from large quantum corrections due to heavy particles/thresholds. One class of such models which solve the hierarchy problem via new TeV-scale dynamics are the composite Higgs (CH) scenarios. Here, the Higgs boson is no longer a fundamental scalar but rather a bound state of a new strong interaction, resolvable only at short distances. Thus, quantum corrections are cutoff at the compositeness scale and the Higgs mass is saturated in the infra-red [1–3], screening it from large corrections. In order to reduce the mass of the Higgs boson to $\mathcal{O}(100\,\text{GeV})$ (vs. other composite resonances which generically have to be in the multi-TeV range) the Higgs also needs to be [4] a pseudo Nambu-Goldstone boson of a spontaneously broken global symmetry of the strong sector $G \to H$. This also has the added benefit that the Higgs potential becomes calculable, since it is radiatively generated by the couplings of the composite sector to the SM, which explicitly break the global symmetry. In particular, the SM fermions generically do not fill complete representations of $G$, hence their interactions with the composite sector will violate the shift symmetry protecting the Higgs mass. The leading source of explicit breaking are usually the interactions responsible for generating the top Yukawa coupling. For reviews see [5,6].

While composite Higgs models provide a very appealing mechanism for generating the Higgs potential dynamically, their minimal realizations (for example where $G = SO(5)$ and $H = SO(4)$ [7]) generically predict a Higgs mass that is too heavy due to the large couplings in the top sector. This in turn requires anomalously light top partners below the generic NP scale to keep the Higgs light by reducing the explicit breaking of the shift symmetry [8–12]. However the recent direct LHC bounds can constrain top partners to be as heavy as $m_T \gtrsim 1.3\,\text{TeV}$ [13–15], which in turn requires the global symmetry breaking scale $f$ to be above the 1 TeV range. Since the tuning needed to obtain a phenomenologically viable minimum in the Higgs potential increases with the scale $f$ in these minimal models, the direct bounds on the top partner masses will start pushing the tuning towards the percent level. In addition to this irreducible tuning of order $\xi \equiv v^2/f^2$, minimal models in which the SM fermions are embedded in fundamental representations of $SO(5)$ also suffer from an extra "double tuning" [16] that considerably aggravates the overall tuning. Moreover, increasing $f$ makes the Higgs more elementary and pushes all the composite resonances beyond the reach of the LHC.

In this article, we present a model that solves both of these problems. For this we will make use of two recently introduced concepts: the soft breaking of the Higgs shift symmetry and the emergent 'maximal symmetry' of the composite sector. The idea of soft breaking [17] is to supplement the SM fermions by additional elementary vector-like fermions to form complete $SO(5)$ multiplets. In this case the source for the $SO(5)$ breaking will be the masses of the vector-like fermions, thus removing the direct link between the top Yukawa coupling and the magnitude of the generated Higgs potential. Maximal symmetry [18,19] is somewhat similar: here the fermions of the composite sector will form complete $SO(5)$ multiplets, and the emerging leftover global $SO(5)'$ symmetry will result in the elimination of the double tuning of the Higgs potential. One can see that these two concepts perfectly complement each other: the latter reduces the absolute size of the tuning while the former tames the dependence on the top partner mass. The model with both soft breaking and maximal symmetry will allow a small tuning with heavy top partner masses, leading to a realistic and minimally tuned vacuum. The model predicts a natural spectrum of resonances, expected to be explorable at the high-luminosity LHC or FCC [20–23]. Since both of these concepts involve complete $SO(5)$ multiplets it also appears very natural to try to combine them. In fact we will show that it is very easy to find an implementation of our model in the context of warped extra dimen-

sions [24, 25]. It simply corresponds to bulk fermions that have $SO(5)$ invariant boundary conditions that are universal on the UV and IR branes. The 5D model is a very simple modification of the canonical holographic MCHM$_5$ which nevertheless automatically yields heavier top partners without requiring large tuning. In the context of the warped implementation we can also easily see that it is not unnatural to keep the additional elementary fermions, needed to obtain the correct SM fermion spectrum, in the right mass range where their effect on reducing the tuning is sizeable. Their lightness is an automatic consequence of the third generation quarks being heavy (hence mainly composite).

This article is organized as follows. We start in Section 2 with a review of the main ingredients, i.e., maximal symmetry, which we discuss here from a spurion perspective, and soft breaking of the Higgs shift symmetry. This sets the stage for the proposed natural CH incarnation with no ultra-light top partners, presented in detail in Section 3 and scrutinized analytically as well as numerically in Section 4, where we also compare the tuning to that of other well-known CH models. To corroborate the naturalness of the setup, in Section 5 we present the five-dimensional (5D) holographic dual of our scenario, where the global symmetry-restoration corresponds to choosing universal boundary conditions for the $SO(5)$ multiplets. In particular, we show how appropriate soft-breaking terms can emerge from fundamental input parameters. Finally, we conclude in Section 6.

## 2  Preliminaries

Before presenting our full model in the next section, it will be useful to review the main ideas behind the crucial ingredients: maximal symmetry and soft breaking. Their combination will lead to interesting synergies in generating a viable Higgs potential, generically parametrized as

$$V(h) = \alpha \sin^2(h/f) + \beta \sin^4(h/f), \tag{1}$$

and inducing a vacuum misalignment angle $\xi = \sin^2(\langle h \rangle / f) = v^2/f^2 = -\alpha/(2\beta)$ which is the key parameter characterizing the tuning and the deviations from the SM predictions.

### 2.1  Key concepts of maximal symmetry

We start by summarizing the key concepts of maximal symmetry [18, 19], considering the $SO(5)/SO(4)$ setup where both chiralities of the SM quarks are embedded in the fundamental **5** of $SO(5)$, which is referred to as MCHM$_5$. Without maximal symmetry, this embedding is known to suffer from the double-tuning problem [16]: at the leading order in the symmetry breaking couplings, the potential (1) contains only one trigonometric function (in particular, $\beta = 0$), thus having only trivial extrema at $h = 0, \pi f$. As a consequence, additional tuning is needed such that next-to-leading terms can allow for the correct EWSB.

This can be understood most easily using a spurion analysis. The embedding of the SM fermions into the **5** of $SO(5)$ is achieved through matrices $\Delta_{q_L, t_R}$ via

$$\bar{q}_L \rightarrow \bar{\psi}_{q_L} = \bar{q}_L \Delta_{q_L}, \quad \bar{t}_R \rightarrow \bar{\psi}_{t_R} = \bar{t}_R \Delta_{t_R}, \tag{2}$$

where the $\Delta$'s are the spurions characterizing the symmetry breaking due to the embedding of the SM fermions into the $SO(5)$ global symmetry even though they do not form complete multiplets. The numerical values of these spurions are

$$\Delta_{q_L} = \frac{1}{\sqrt{2}} \begin{pmatrix} 0 & 0 & 1 & -i & 0 \\ 1 & i & 0 & 0 & 0 \end{pmatrix}, \quad \Delta_{t_R} = \begin{pmatrix} 0 & 0 & 0 & 0 & i \end{pmatrix}. \tag{3}$$

The elementary SM fermions $\psi_{q_L}$ and $\psi_{t_R}$ mix linearly with the composite resonances in the low energy effective Lagrangian, leading to masses for the SM fermions after EWSB, providing a simple implementation of the partial compositeness paradigm [7,8,26,27]. Since the spurions connect different symmetry groups they will have mixed indices: the columns transform under the elementary (SM-like) $SU(2)_L^0$ symmetry and the rows under $SO(5)$:

$$\Delta_{q_L} \sim \left[SU(2)_L^0 \times U(1)_Y^0\right] \times SO(5), \quad \Delta_{t_R} \sim U(1)_Y^0 \times SO(5). \tag{4}$$

The spurions fully encode the effects of the explicit breaking due to using incomplete multiplets, which in turn controls the potential for the Higgs as a pseudo Nambu-Goldstone boson.[1] In practice, one first treats the spurions as dynamical fields and identifies the transformations that would formally restore $SO(5)$ as a true symmetry. Physical quantities, such as the Higgs potential, need to be invariant when *all* the fields, including the spurions, are transformed. This constrains the possible combinations of fields that can enter the Higgs potential. Once all the invariants are constructed, the spurions can be set to their actual form in (3), that can be regarded as the vacuum expectation value of the corresponding field.

The relevant symmetries here are the elementary $SU(2)_L^0 \times U(1)_Y^0$ symmetry and the $SO(5)$ symmetry of the composite sector. As we shall see, the Higgs boson only transforms under $SO(5)$.[2] This forces the spurions to always appear in hermitian conjugate pairs with the elementary indices (labeled by Greek letters) contracted among themselves, and complex conjugation ensuring $U(1)_Y^0$ invariance. Thus the spurions can enter the Higgs potential only through the following combinations [9]:

$$(\Gamma_L)_{IJ} \equiv (\Delta_{q_L}^*)_I^\alpha (\Delta_{q_L})_{\alpha J}, \quad (\Gamma_R)_{IJ} \equiv (\Delta_{t_R}^*)_I (\Delta_{t_R})_J, \tag{5}$$

with $\alpha = 1,2$ the $SU(2)_L^0$ indices, while $I,J = 1,\ldots,5$ are $SO(5)$ indices. As the latter are the only free indices left, the $\Gamma_{L,R}$ spurions only transform under $SO(5)$:

$$\Gamma_{L,R} \to g\,\Gamma_{L,R}\,g^\dagger, \quad g \in SO(5). \tag{6}$$

Because of its Nambu-Goldstone nature, the Higgs field $h^{\hat{a}}$ belongs to the $SO(5)/SO(4)$ coset and always appears through the Goldstone matrix $U$,

$$U = \exp\left(\frac{ih^{\hat{a}}T^{\hat{a}}}{f}\right), \tag{7}$$

which transforms non-linearly under $SO(5)$:

$$U \to g^\dagger U h^\dagger(h^a, g), \quad g \in SO(5),\ h \in SO(4). \tag{8}$$

However, for symmetric cosets as $SO(5)$, there exists an automorphism $V$ called Higgs parity, with $V T^A V^\dagger = s_A T^A$, where $s_A = +/-$ for the unbroken/broken generators $A = a/\hat{a}$, that can be used to define a new matrix $\Sigma$ with linear transformation properties [18]:

$$\Sigma = U^2 V \to g\,\Sigma\,g^\dagger, \quad g \in SO(5). \tag{9}$$

The $\Gamma_{L,R}$ spurions and the linear Goldstone matrix $\Sigma$ defined in (5) and (9), respectively, are the ingredients needed to investigate the fermion contribution to the Higgs potential. Higher orders in perturbation theory correspond to larger number of insertions of the $\Gamma$ spurions

---

[1]As a consequence, each term in the potential must contain at least one spurion.

[2]The electroweak group corresponds to the gauged diagonal $SU(2)_L \times U(1)_Y$ subgroup of the elementary and composite global symmetries, and we omit an additional $U(1)_X$ factor that is not crucial here [6,12].

(keeping the same order in the loop expansion). The leading order corresponds to one spurion insertion. One finds that at this order there are only two different invariants:[3]

$$V_{\text{LO}}(h) = c_L \text{Tr}(\Sigma\,\Gamma_L) + c_R \text{Tr}(\Sigma\,\Gamma_R) = (2c_R - c_L)\sin^2(h/f), \tag{10}$$

not allowing for non-trivial extrema. One thus has to rely on a cancellation with a term that is formally sub-leading to generate a realistic minimum.

Let us now show how maximal symmetry [18] solves this issue. In CH models composite fermionic resonances will appear in the spectrum which in general do not need to fill complete $SO(5)$ representations (but they always have to obey the unbroken SO(4) global symmetry). The assumption of maximal symmetry is that such resonances nevertheless still come in complete $SO(5)$ multiplets. For generic values of the resonance masses, the residual symmetry is still only $SO(4)$. However, if the masses were neglected, we see that the original $SO(5)$ would be actually doubled to the chiral group $SO(5)_L \times SO(5)_R$. Interestingly, there exists a choice of resonance masses that exhibits a residual symmetry larger than $SO(4)$, which is referred to as maximal symmetry. Technically, this can be defined as the largest symmetry group that can be preserved by turning on non-zero masses for the composite states that still gives a non-vanishing Higgs potential. In practice, maximal symmetry turns out to be the $SO(5)'$ subgroup of $SO(5)_L \times SO(5)_R$ that satisfies

$$g_L^{\prime\,\dagger} V g_R' = V, \quad g_L' \in SO(5)_L, \quad g_R' \in SO(5)_R, \tag{11}$$

where $V$ is the Higgs parity operator introduced above (9). Another possibility would be the $SO(5)_V$ subgroup defined by $g_L \mathbb{1} g_R^\dagger = \mathbb{1}$, which however would make the Higgs an exact Nambu-Goldstone boson. When $SO(5)'$ is promoted to a symmetry of the theory, more insertions of the spurions $\Gamma_{L,R}$ are needed in order to generate a potential. Indeed, possible contributions are now more constrained, as they need to be invariant not only under $SU(2)_L^0 \times U(1)_Y^0$ and $SO(5)$ as before, but also under $SO(5)'$.

To see this explicitly, we first have to identify the transformation properties of $\Gamma_{L,R}$ under $SO(5)'$. To this end, it is convenient to factor the $U$ matrix together with the fermion fields, $\Psi_{L,R}$, such that a generic chiral transformation reads

$$(U\Psi_L) \to g_L (U\Psi_L), \quad (U\Psi_R) \to g_R (U\Psi_R). \tag{12}$$

The way $\Gamma_{L,R}$ transform under maximal symmetry turns out to be a simple 'chiral' generalization of (6):

$$\Gamma_L \to g_R\,\Gamma_L\,g_R^\dagger, \quad \Gamma_R \to g_L\,\Gamma_R\,g_L^\dagger. \tag{13}$$

The only twist is that due to partial compositeness $q_L$ couples to the right-handed composites that by definition transform with $SO(5)_R$, and similarly for $t_R$. The matrices $g_{L,R}$ are related to the ones in (11) as $g_{L,R}' = U^\dagger g_{L,R} U$, and the condition (11) has the equivalent form

$$g_L^\dagger \Sigma g_R = \Sigma. \tag{14}$$

We can perform a spurion analysis by constructing operators containing $\Gamma_{L,R}$ and $\Sigma$ that are formally invariant under (13) making use of (14). Alternatively, we may assign spurious transformation properties to the $\Sigma$ matrix itself under $SO(5)'$ that incorporate the defining property of maximal symmetry,

$$\Sigma \to g_L \Sigma g_R^\dagger, \tag{15}$$

and require a given term of the potential to be formally invariant under the simultaneous action of (13) and (15).

---

[3]The coefficients $c_{L,R}$ can be fixed in an explicit calculation.

Table 1: Transformation properties of the spurions $\Gamma_{L,R}$ defined in (5) and the linear Goldstone matrix $\Sigma$ under the global symmetry group, $SO(5)$, and maximal symmetry, $SO(5)'$. The $\Sigma$ transformation under $SO(5)'$ should be regarded as spurious, see discussion around Eq. (15).

|  | $SO(5)$ | $SO(5)'$ |
|---|---|---|
| $\Gamma_L$ | $\Gamma_L \to g\,\Gamma_L\,g^\dagger$ | $\Gamma_L \to g_R\,\Gamma_L\,g_R^\dagger$ |
| $\Gamma_R$ | $\Gamma_R \to g\,\Gamma_R\,g^\dagger$ | $\Gamma_R \to g_L\,\Gamma_R\,g_L^\dagger$ |
| $\Sigma$ | $\Sigma \to g\,\Sigma\,g^\dagger$ | $\Sigma \to g_L\,\Sigma\,g_R^\dagger$ |

Given this set of rules it is apparent that both terms in (10) are forbidden by maximal symmetry. In fact, the leading contribution to the potential requires at least two spurions to appear simultaneously, i.e.,

$$V_{\mathrm{LO}}(h) = c_{\mathrm{LR}}\mathrm{Tr}(\Sigma\,\Gamma_L\,\Sigma^\dagger\,\Gamma_R) = 2\,c_{\mathrm{LR}}\sin^2(h/f)\cos^2(h/f). \tag{16}$$

A summary of the transformation properties that we have used to derive (10) and (16) can be found in Table 1. The main difference compared to (10) is that now a non-trivial minimum occurs already at the leading order, thus solving the double-tuning problem. However, the minimum arising from (16) is still rather special, as it corresponds to $\alpha = -\beta$ and thus implies

$$\xi \equiv \sin^2(\langle h\rangle/f) = 0.5, \tag{17}$$

independently of any choice of parameters. Such a value of $\xi$ is by now excluded experimentally – but in principle the gauge sector can come to rescue since it contributes to the Higgs potential as well and, with a small degree of accidental cancellation, can help misaligning the vacuum in the right way [18]. However it would be interesting to avoid this and solve the double-tuning issue without restriction.

The reason of this sharp prediction for $\xi$ is the appearance of a discrete exchange symmetry

$$\sin(h/f) \longleftrightarrow -\cos(h/f). \tag{18}$$

For a symmetric coset, this trigonometric parity is always a symmetry related to the existence of the automorphism $V$. Requiring maximal symmetry thus ensures that it remains a symmetry of the whole theory. Trigonometric parity in fact plays a crucial role in forbidding the linear terms in (10) and hence reduces the corrections to the Higgs mass, similarly to what happens in Twin Higgs constructions [28, 29]. [4]

We will see that once we also introduce soft-breaking, trigonometric parity is broken in a way that avoids the unwanted prediction $\xi = 0.5$ already in the fermion sector, but at the same time preserves the structure in (16) solving the double-tuning problem. The added advantage of combining maximal symmetry with the soft-breaking mechanism will be to allow for heavier partners while maintaining a light Higgs without further increasing $f$. This can lead to a natural spectrum of resonances above 2 TeV as we will see in detail in Sec. 3.

## 2.2 The soft-breaking setup

Let us now recall the main idea behind the soft-breaking setup proposed in [17]. Here, the key concept is to enhance the symmetry of the couplings responsible for partial compositeness

---

[4]Maximal symmetry is an emerging symmetry of the composite sector, and can not be enforced by imposing a symmetry structure of the UV theory. It is rather the consequence of the specific dynamics responsible for the composite sector. Within an effective theory approach it is simply an additional assumption on the structure of the composite sector.

by completing the SM fermions to full representations of the global symmetry. This requires new vector-like quarks in the theory, which in turn explicitly break the shift symmetry via their soft mass terms. In order for these new degrees of freedom to significantly affect the Higgs potential, their mass needs to be around the TeV scale.[5]

Although a UV completion of composite Higgs models with partial compositeness is notoriously challenging in 4D – especially for the minimal coset $SO(5)/SO(4)$ – the soft-breaking setup may be thought of as a microscopic theory in which the fundamental interactions between the elementary fields and the fundamental constituents of the strong sector respect a certain global symmetry of the theory – $SO(5)$ in this case – which is broken only via soft mass terms in the elementary sector.

The main phenomenological advantage of this setup is that one can raise the top partner masses while keeping the Higgs mass fixed *without* having to raise $f$, unlike in the MCHM$_5$ and its maximally symmetric version. It was found in [17] that in this setup $\beta$ from (1) (which fixes the Higgs mass) is generically reduced compared to the MCHM$_5$. The quadratic term $\alpha$ however remains almost unchanged, implying that the overall tuning is eventually similar to that in the MCHM$_5$. Nonetheless, the crucial difference is that the tuning needed to achieve heavier top partners in the soft-breaking setup is not irreducible, since it is not coming from a very small misalignment angle (for constant top partner masses $f$ can be smaller). Thus, it can be drastically cut down whenever other ingredients are added to the model. Maximal symmetry is then the ideal candidate, as it provides the crucial connection between $\beta$ and $\alpha$ through trigonometric parity. As we will see in the next section, the outcome is then a fully softened Higgs potential.

To illustrate these points in detail, let us focus on the softended MCHM$_5$ with minimal fermion embeddings, dubbed sMCHM$_5$, and investigate the compatibility of maximal symmetry with this minimal proposal of Ref. [17], employing three new elementary fermions $v, w$ and $s$. The SM quarks $q_L$ and $t_R$ are part of full (elementary) $\mathbf{5}$ representations of $SO(5)$, $\psi_L^t$ and $\psi_R^t$ with

$$\psi_L^t = \Delta_{q_L}^\dagger q_L + \Delta_w^\dagger w_L + \Delta_s^\dagger s_L, \quad \psi_R^t = \Delta_{t_R}^\dagger t_R + \Delta_w^\dagger w_R + \Delta_v^\dagger v_R, \tag{19}$$

where the new spurions are given by

$$\Delta_s = \Delta_{t_R}, \quad \Delta_v = \Delta_{q_L}, \quad \Delta_w = \frac{1}{\sqrt{2}}\begin{pmatrix} 1 & -i & 0 & 0 & 0 \\ 0 & 0 & 1 & i & 0 \end{pmatrix}. \tag{20}$$

As emphasized before this amounts to restoring complete $SO(5)$ multiplets by reintroducing the missing components:

$$\Delta_{q_L}^\dagger q_L = \frac{1}{\sqrt{2}}\begin{pmatrix} b_L \\ -ib_L \\ t_L \\ it_L \\ 0 \end{pmatrix} \rightarrow \psi_L^t = \frac{1}{\sqrt{2}}\begin{pmatrix} b_L - w_L^1 \\ -ib_L - iw_L^1 \\ t_L + w_L^2 \\ it_L - iw_L^2 \\ -i\sqrt{2}s_L \end{pmatrix}, \tag{21}$$

and

$$\Delta_{t_R}^\dagger t_R = \begin{pmatrix} 0 \\ 0 \\ 0 \\ 0 \\ -it_R \end{pmatrix} \rightarrow \psi_R^t = \frac{1}{\sqrt{2}}\begin{pmatrix} v_R^2 - w_R^1 \\ -iv_R^2 - iw_R^1 \\ v_R^1 + w_R^2 \\ iv_R^1 - iw_R^2 \\ -i\sqrt{2}t_R \end{pmatrix}. \tag{22}$$

---

[5]Since these are elementary degrees of freedom, the requirement of TeV scale masses could introduce a coincidence problem which we will discuss in Sec. 5.

The most general set of masses and mixings between the SM quarks and the new vector-like fermions is given by

$$
\begin{aligned}
-\mathcal{L}_{\text{el}} =\ & m_w(\bar{w}_L w_R + \bar{w}_R w_L) + m_v(\bar{v}_L v_R + \bar{v}_R v_L) + m_s(\bar{s}_L s_R + \bar{s}_R s_L) \\
& + (\delta_1 \bar{s}_L t_R + \delta_2 \bar{q}_L v_R + \text{h.c.}),
\end{aligned}
\tag{23}
$$

whereas the partial compositeness Lagrangian reads

$$
\begin{aligned}
-\mathcal{L}_{\text{mass}} =\ & m_{\mathbf{4}} \bar{Q}_L Q_R + m_{\mathbf{1}} \bar{\tilde{T}}_L \tilde{T}_R \\
& + y_L f\, \bar{\psi}^t_{LI} \left( a_L U_{Ii} Q^i_R + b_L U_{I5} \tilde{T}_R \right) \\
& + y_R f\, \bar{\psi}^t_{RI} \left( a_R U_{Ii} Q^i_L + b_R U_{I5} \tilde{T}_L \right) + \text{h.c.}
\end{aligned}
\tag{24}
$$

This Lagrangian generically describes the interactions of the elementary fields $\psi^t_{L,R}$ with the lightest resonances $Q$ and $\tilde{T}$ of the strong sector, transforming as $Q \sim \mathbf{4}$ and $\tilde{T} \sim \mathbf{1}$ under $SO(4)$. The model described by (23) and (24) is the one discussed in [17]. Due to the use of full $SO(5)$ multiplets in (21) and (22), all the explicit breaking of the $SO(5)$ global symmetry is contained in (23) corresponding to the masses of the new vector-like fermions in addition to possible mixing terms with the SM quarks.

As emphasized before, the main advantage of this setup is that the direct link between the top-Yukawa and the Higgs potential is removed, and it becomes possible to raise the top partner masses at constant $f$, while keeping the Higgs fixed at 125 GeV. This effect can already be captured by looking at the case in which the singlet $s$ is much lighter than the other vector-like fermions. The only new parameter with respect to the MCHM$_5$ is the singlet mass, $m_s$, and one can obtain simple analytical formulae for the lightest top-partner mass, $m_T$. In [17], it was found that the latter is given by

$$
m_T \simeq 2.2 \frac{m_h}{m_t} \frac{1 - \epsilon/4}{\sqrt{\epsilon}} f,
\tag{25}
$$

where we have taken $m_{\mathbf{4}} = -m_{\mathbf{1}} = M$ for concreteness (although the overall behavior does not depend on this choice), and

$$
\epsilon \equiv 1 - \frac{M}{m_s},
\tag{26}
$$

controls the impact of the singlet state. If $m_s$ is much above the mass-scale of the composites, $m_s \gg M$, one has $\epsilon \approx 1$, and in that limit all the results of the MCHM$_5$ are recovered, in particular $m_T \simeq 1.1$ TeV with $f = 800$ GeV. Conversely, if $m_s$ is comparable to $M$, one has $\epsilon < 1$ and (25) always yields heavier top partners. For instance, the case of $m_s \simeq 2M$ ($\epsilon = 0.5$) already implies $m_T \simeq 1.8$ TeV for the same value of $f$. However, as mentioned above, the overall tuning of this model is still rather large, because the setup still inherits the typical double tuning of the MCHM$_5$.

As a warm-up let us try to implement maximal symmetry in (24) in the most naive way. This would simply correspond to setting $m_{\mathbf{4}} = -m_{\mathbf{1}}$, $a_L = b_L$ and $a_R = b_R$. In this case, the resonances *also* always appear as full multiplets in the $\mathbf{5}$ of $SO(5)$, $\Psi \equiv (Q, \tilde{T})$. As discussed in Sec. 2.1, under the $SO(5)'$ transformations the chiral components of $\Psi$ transform with $g_L \in SO(5)_L$ and $g_R \in SO(5)_R$ as

$$
U\Psi_L \to g_L (U\Psi_L), \quad U\Psi_R \to g_R (U\Psi_R),
\tag{27}
$$

where we have used the fermions dressed by the Goldstone matrix $U$ in the definition of $SO(5)'$ as in [18]. As before, the contributions to the Higgs potential must be proportional to the terms that explicitly break the $SO(5)$ symmetry. In the soft-breaking setup, these are given in (23).

For the moment, let us focus on the contribution from terms involving the new fermion $w$, whose vector-like mass $m_w$ breaks the $SO(5)$. To identify the proper transformation property of this term it is useful to rewrite $m_w \bar{w}_L w_R$ in terms of the full multiplets $\psi_{L,R}^t$ as

$$m_w \bar{w}_L w_R = \bar{\psi}_L^t \Gamma_w \psi_R^t, \tag{28}$$

where $\Gamma_w$ is the corresponding spurion that encodes the explicit breaking:

$$\Gamma_w = \frac{1}{2} m_w \begin{pmatrix} 1 & -i & 0 & 0 & 0 \\ i & 1 & 0 & 0 & 0 \\ 0 & 0 & 1 & i & 0 \\ 0 & 0 & -i & 1 & 0 \\ 0 & 0 & 0 & 0 & 0 \end{pmatrix}. \tag{29}$$

In order to derive how the elementary multiplets $\psi_{L,R}^t$ transform under $SO(5)'$, we perform the transformation in (27) and demand that the full Lagrangian is invariant. One can see that the elementary fields need to transform as $\psi_L^t \to g_R \psi_L^t$ and $\psi_R^t \to g_L \psi_R^t$, and this implies that $\Gamma_w$ transforms as

$$\Gamma_w \to g_R \Gamma_w g_L^\dagger. \tag{30}$$

In fact, now there is still one invariant using (30)

$$V_{\text{LO}}(h) = c_w \text{Tr}(\Sigma \Gamma_w) \propto m_w \sin^2(h/f), \tag{31}$$

which is allowed in the Higgs potential. Since there is no $m_w \cos^2(h/f)$ balancing (31), we see that trigonometric parity is badly broken leading again to double-tuning as discussed in Sec. 2.1. The same type of contribution is found considering the $\delta_{1,2}$ terms in (23). In general, whenever $\psi_L^t$ and $\psi_R^t$ have a direct interaction term as in (28), the trigonometric parity is badly broken and double-tuning is reintroduced.[6]

We conclude that the simplest realization of soft-breaking in the MCHM$_5$ specified by the new vector-like fermions in (19) is not directly compatible with maximal symmetry. However in the next section, we show that a slight change in the embedding (19) will allow us to successfully combine maximal symmetry and soft-breaking, which will lead to an increase in the mass of the top partners with minimal tuning and to the disappearance of the unwanted prediction for the misalignment angle.

## 3 Successfully combining soft breaking and maximal symmetry

We are now ready to introduce our simple model in which maximal symmetry and soft-breaking are successfully combined, resulting in a composite resonance spectrum naturally above the LHC bounds. As we have seen, the only obstacle was the mass of the vector-like fermion $w$ which badly broke maximal symmetry and reintroduced the double-tuning. We will now show that there is a simple way to avoid the double tuning. All we need to do is to further split the vector-like fermion $w$ into two: rather than marrying up $w_L$ appearing in $\psi_L$ directly with $w_R$ appearing in $\psi_R$, we introduce separate partners for these two $w$'s. Hence our embedding will be

$$\psi_L^t = \Delta_{q_L}^\dagger q_L + \Delta_{w_1}^\dagger w_{1L} + \Delta_s^\dagger s_L, \quad \psi_R^t = \Delta_{t_R}^\dagger t_R + \Delta_{w_2}^\dagger w_{2R} + \Delta_v^\dagger v_R, \tag{32}$$

---

[6]Clearly, trigonometric parity is restored if $s, v, w$ become infinitely heavy, since the MCHM$_5$ is effectively recovered and one is left with the standard maximally-symmetric model where heavier top partners can only appear at the price of raising $f$.

where $\Delta_{w_1} = \Delta_{w_2} = \Delta_w$. For the mass terms of the elementary fields we will take a simple modification of (23):

$$-\mathcal{L}_{\text{el}} = m_{w_1}(\bar{w}_{1L}w_{1R} + \bar{w}_{1R}w_{1L}) + m_{w_2}(\bar{w}_{2L}w_{2R} + \bar{w}_{2R}w_{2L}) \\ + m_\nu(\bar{\nu}_L\nu_R + \bar{\nu}_R\nu_L) + m_s(\bar{s}_Ls_R + \bar{s}_Rs_L). \tag{33}$$

Note that there are additional mixing terms that would be allowed by the SM gauge symmetries. We will discuss these below in (37).

It is convenient to collect the chiralities that do not enter either of $\psi^t_{L,R}$ in two multiplets,

$$\eta_R \equiv (w_{1R}, s_R), \quad \xi_L \equiv (w_{2L}, \nu_L). \tag{34}$$

The full Lagrangian of our model in terms of these fields is then

$$-\mathcal{L} = \bar{\psi}^t_L M^\dagger_R \eta_R + \bar{\psi}^t_R M^\dagger_L \xi_L + y_L f \bar{\psi}^t_L U \Psi_R + y_R f \bar{\psi}^t_R U \Psi_L + M \bar{\Psi}_L V \Psi_R + \text{h.c.}, \tag{35}$$

where $V$ is the usual Higgs parity $V = \text{diag}(1, 1, 1, \sigma_3)$, and for maximal symmetry we choose $M = m_4 = -m_1$, see below Eq. (24). The first two terms correspond to a compact way of writing (33) via matrices accounting for the masses of the elementary vector-like fermions:

$$M^\dagger_R = \frac{1}{\sqrt{2}}\begin{pmatrix} m_{w_1} & 0 & 0 \\ im_{w_1} & 0 & 0 \\ 0 & m_{w_1} & 0 \\ 0 & -im_{w_1} & 0 \\ 0 & 0 & im_s \end{pmatrix}, \quad M^\dagger_L = \frac{1}{\sqrt{2}}\begin{pmatrix} m_{w_2} & 0 & 0 & m_\nu \\ im_{w_2} & 0 & 0 & -im_\nu \\ 0 & m_{w_2} & m_\nu & 0 \\ 0 & -im_{w_2} & im_\nu & 0 \\ 0 & 0 & 0 & 0 \end{pmatrix}, \tag{36}$$

where the columns correspond to the $SO(5)$ indices while the rows to the $SU(3)^0$ vs. $SU(4)^0$ global symmetries of the kinetic terms of the $\eta_R$ and $\xi_L$ multiplets.

As we mentioned before, there are three more mixing terms between the elementary fields that would be allowed, which are given by

$$-\mathcal{L}_{\text{odd}} = \delta_1 \bar{s}_L t_R + \delta_2 \bar{q}_L \nu_R + \delta_{12} \bar{w}_1 w_2 + \text{h.c.} \\ \equiv \bar{\psi}^t_L \Delta(\delta_1, \delta_2) \psi^t_R + \bar{\xi}_L \Delta'(\delta_{12}) \eta_R + \text{h.c.}, \tag{37}$$

where the last line defines the spurions $\Delta(\delta_1, \delta_2)$ and $\Delta'(\delta_{12})$. Based on the discussion in the previous section, it is clear that a non-zero value for any of the $\delta$'s in (37) would be incompatible with maximal symmetry and reintroduce the double-tuning; we thus require $\delta_1 = \delta_2 = \delta_{12} = 0$. This can be easily achieved by introducing a $Z_2$ symmetry under which the parities of $\psi^t_L$ and $\eta_R$ are opposite to the parities of $\psi^t_R$ and $\xi_L$, for example $\psi^t_L, \eta_R : +$ and $\psi^t_R, \xi_L : -$. When considering the whole Lagrangian in (35), the $Z_2$ is broken softly by the the composite mass $M$. In the 5D picture (see Sec. 5), this will have a very nice interpretation corresponding to a $Z_2$ symmetry that is only broken on the IR brane.

Let us now investigate the key properties of our main model defined in (35), regarding double-tuning and trigonometric parity, by using spurion analysis. For this we need to derive the transformation properties of the $M_{R,L}$ spurions in (36). First, as long as the electroweak gauge interactions are neglected, when $M_{R,L} = 0$ the fields $\eta_R$ and $\xi_L$ are "free" and exhibit a large symmetry of their kinetic terms, i.e. $SU(3)^0 \times SU(4)^0$ (notice that $\eta_R$ is a triplet and $\xi_L$ a fourplet). This large symmetry extends the $SU(2)^0_L \times U(1)^0_Y$ discussed above Eq. (5), and similarly implies that $M_{L,R}$ enter the potential only through the combinations

$$\Gamma_R \equiv M^\dagger_R M_R, \quad \Gamma_L \equiv M^\dagger_L M_L, \tag{38}$$

which transform under $SO(5)'$ similarly to (13) (except with $L \leftrightarrow R$):

$$\Gamma_R \to g_R \Gamma_R g_R^\dagger, \quad \Gamma_L \to g_L \Gamma_L g_L^\dagger. \tag{39}$$

We note that, unlike in Sec. 2.1, higher orders in the expansion parameter $y_{L,R}/g_*$ (with $g_*$ the typical interaction strength of the composite states) do not correspond to more insertions of the spurions $\Gamma_{L,R}$, which only depend on the vector-like masses. The leading order Higgs potential is rather determined by the least number of $\Sigma$ insertions, since the Higgs only enters through the Goldstone matrix that always appears together with $y_{L,R}$ in (35). Due to (39), the leading contribution requires two $\Sigma$'s to appear simultaneously and its structure is fixed as:

$$V_{\text{LO}}(h) = c_{LR} \sum_{i,j=1}^\infty a_{ij} \text{Tr}(\Sigma^\dagger \Gamma_L^i \Sigma \Gamma_R^j), \tag{40}$$

where $i, j$ are arbitrary powers for the $\Gamma_{L,R}$ matrices, for which (40) is still formally invariant under $SO(5)'$, and the coefficients $a_{ij}$ can be determined from explicit calculation. Next-to-leading terms in the potential correspond to more insertions of $\Sigma$. On the other hand, since the elementary vector-like masses are not necessarily small, all powers of $\Gamma_{L,R}$ can in principle contribute. In order to illustrate the effects of the terms in (40), we explicitly evaluate the first one corresponding to $i = j = 1$:

$$\begin{aligned}
V_{\text{LO}}^{(1,1)}(h) \propto &\left( m_s^2 m_v^2 + m_s^2 m_{w2}^2 - 2m_{w1}^2 m_{w2}^2 \right) \sin^2(h/f) \\
&+ (m_v^2 m_{w1}^2 - m_s^2 m_v^2 + m_{w1}^2 m_{w2}^2 - m_s^2 m_{w2}^2) \sin^4(h/f),
\end{aligned} \tag{41}$$

which would give $\xi = 0.1$ for instance for $m_s = 2.4$ TeV, $m_{w1} = 3$ TeV, $m_{w2} = 4$ TeV and $m_v = 5$ TeV (although the actual value of $\xi$ is expected to change when also including the terms with $i, j > 1$).

We thus conclude that the soft MCHM$_5$ with maximal symmetry specified in (35) is free from double-tuning, because the structure in (40) is rich enough to provide a non-trivial minimum for EWSB at the leading order. Moreover, we notice that $\xi$ is not constrained to be $\xi = 0.5$ as it was found for the leading-order potential in (16). As we have seen, this prediction for the misalignment $\xi$ is controlled by trigonometric parity and we can ask what is its fate in the soft-breaking setup. While a detailed analysis is presented in App. A, here we just give the result, which is that trigonometric parity is always broken in the fermion sector of the soft-breaking setup with the exception of some particular values for the vector-like fermion masses:

$$s_h \leftrightarrow -c_h \text{ is unbroken} \Rightarrow \xi = 0.5 \quad \Leftrightarrow \quad (m_v^2 - m_{w_2}^2)m_{w_1}^2 = 0. \tag{42}$$

We then conclude that, for generic values of the vector-like masses, the unwanted prediction $\xi = 0.5$ can be avoided *without* reintroducing the double tuning, since at generic points trigonometric parity is broken in a controlled way by the vector-like masses.

In the next section, we will provide a quantitative analysis of the potential in (40) and calculate the tuning in our model, which will turn out to be natural also with top partner masses above 2 TeV.

## 4 Heavy top partners with minimal tuning

Next we present the quantitative results for our model and calculate the amount of tuning needed in order to achieve correct EWSB and heavy top and gauge partners above the LHC

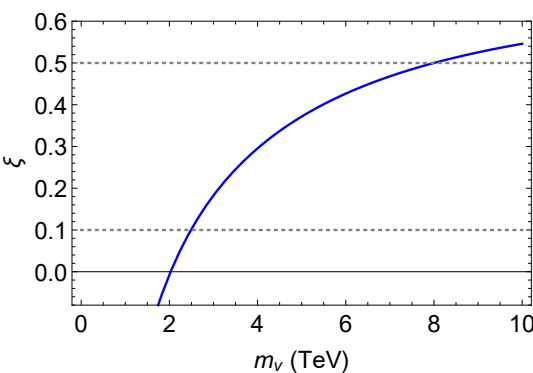

Figure 1: The value of $\xi \equiv \sin^2(\langle h \rangle / f) = -\alpha/2\beta$ as a function of $m_v$ for $M = 2.6$ TeV, $m_{w_1} = m_{w_2} = 8$ TeV, $m_s = 2.4$ TeV.

bounds. Using the standard parametrization of the potential (1), we find at leading order in $y_{L,R}$

$$\alpha + \beta = -C \int_0^\infty p^3 \, dp \, \frac{M^2 m_{w_1}^2 (m_v^2 - m_{w_2}^2)}{(M^2 + p^2)^2 (m_{w_1}^2 + p^2)(m_{w_2}^2 + p^2)(m_v^2 + p^2)}, \tag{43}$$

and

$$\beta = C \int_0^\infty p^3 dp \, \frac{M^2 (2m_v^2 m_{w_2}^2 + (m_v^2 + m_{w_2}^2)p^2)(m_s^2(m_{w_1}^2 + 2p^2) - m_{w_1}^2 p^2)}{p^2(p^2 + M^2)^2(p^2 + m_s^2)(p^2 + m_v^2)^2(p^2 + m_{w_1}^2)(p^2 + m_{w_2}^2)}, \tag{44}$$

where $C = \frac{2N_c}{8\pi^2} y_L^2 y_R^2 f^4$. In particular, since $\xi = -\alpha/(2\beta)$, one can see that the point corresponding to unbroken trigonometric parity $\xi = 0.5$ is realized if the integrand in (43) vanishes, which happens when $m_{w_1}^2 (m_v^2 - m_{w_2}^2) = 0$, in agreement with (42).

The result for $\xi$ is shown in Fig. 1 as a function of $m_v$, fixing the other parameters such that the Higgs mass and the top mass are correctly reproduced at the point $\xi = 0.1$. As we can see, the curve hits $\xi = 0.5$ at $m_v = m_{w_2}$ and $\xi$ is slowly varying with $m_v$, such that getting down to $\xi = 0.1$ does not require significant tuning.

In addition to the terms from the top sector (43) and (44), the potential also contains a contribution from the gauge sector. It mainly affects $\alpha$, and we will take this into account by adding the following term [30]:

$$\alpha_g = \frac{9}{64\pi^2} g^2 f^2 m_\rho^2, \tag{45}$$

where $m_\rho$ is the mass of the spin-1 vector resonance $\rho$.

In order to quantitatively estimate the tuning of the theory $\Delta$, we adopt the Barbieri-Giudice measure [31]

$$\Delta = \max\{|\Delta_i|\}, \quad \Delta_i = \frac{2x_i c_h^2}{s_h m_h^2 f^2} \frac{\partial^2 V}{\partial x_i \partial s_h}, \tag{46}$$

where $s_h \equiv \sin(h/f)$ and similarly for $c_h$, and the independent variables $x_i$ are

$$x_i = \{y_L, y_R, f, m_\rho, M, m_s, m_v, m_{w_1}, m_{w_2}\}. \tag{47}$$

Before computing the tuning in our model, let us briefly review the tuning in various other incarnations of composite Higgs models. In the standard MCHM$_5$, the tuning has been estimated in Ref. [16] as

$$\Delta_5 \simeq \frac{1}{\xi} \times 20 \times \left(\frac{g_*}{5}\right)^2 \simeq \frac{f^2}{v^2} \times 10, \tag{48}$$

where the Higgs mass is fixed at $m_h = 125$ GeV and we have taken $g_* \simeq 3.6$ for concreteness. The resulting extra factor of 10 on top of the irreducible tuning $f^2/v^2$ corresponds to the double-tuning extensively discussed above.

For the maximally-symmetric version of the model, the double-tuning is removed and the tuning is reduced to [18]

$$\Delta_5^{\text{max sym}} \simeq \frac{1}{\xi} - 2 \simeq \frac{f^2}{v^2} - 2. \tag{49}$$

In both models, the top partner and Higgs masses are related [9] via

$$m_h \simeq 130 \, \frac{m_T}{1.4 f} \, \text{GeV}, \tag{50}$$

where $m_T$ is the mass scale of the lightest top partner, which gives $f \simeq 0.75 \, m_T$ for $m_h = 125$ GeV. Thus, the tuning in (48) can be expressed as a function of $m_T$ as

$$\Delta_5 \simeq 90 \left(\frac{m_T}{1 \, \text{TeV}}\right)^2, \tag{51}$$

whereas for maximal symmetry, Eq. (49) leads to the expression

$$\Delta_5^{\text{max sym}} \simeq 9 \left(\frac{m_T}{1 \, \text{TeV}}\right)^2. \tag{52}$$

In the original incarnation of soft breaking, the sMCHM$_5$ [17], the tuning as a function of the lightest partner, $m_T$, is expected to roughly follow the estimate for the MCHM$_5$, Eq. (51). This can be understood by first noticing that the sMCHM$_5$ still suffers from double tuning. Furthermore, although heavier top partners with soft breaking are compatible with smaller $f \simeq 800$ GeV, reducing $\beta$ with the help of the vector-like masses as in Eq. (25) to keep the Higgs light requires extra cancellations in $\alpha$ in order to reproduce the correct misalignment, $\xi = -\alpha/(2\beta) \simeq 0.1$. The conservative estimate of unchanged $\alpha$ in the soft breaking setup, together with Eq. (25), eventually leads to a similar dependence on $m_T$ and no significant reduction in the overall tuning.

This picture changes when combining soft breaking and maximal symmetry.[7] In order to estimate the tuning in our new model, Eq. (35), we start again by considering the basic expression for the tuning in the maximally symmetric MCHM$_5$, Eq. (49). The crucial difference, however, is that raising the top partner mass will not require a larger $f$ any more, thus avoiding the quadratic growth with $m_T$ in Eq. (52). Moreover, $\alpha$ and $\beta$ are now connected through trigonometric parity and the softening due to the vector-like masses simultaneously applies to the whole potential. Therefore, as a first approximation, we expect the tuning to be actually given by (49) with $f = 800$ GeV, independently of the top partner masses:

$$\Delta \simeq \frac{1}{\xi} - 2 \simeq 8. \tag{53}$$

Of course, the relation above cannot hold for arbitrarily heavy top partners and will start getting non-negligible corrections above some critical value of $m_T$ due to the fact that one cannot keep raising $m_T$ while holding $f$ and $m_h$ fixed, unless the vector-like masses are pushed

---

[7]We thank Kaustubh Agashe for clarifying discussions on this topic.

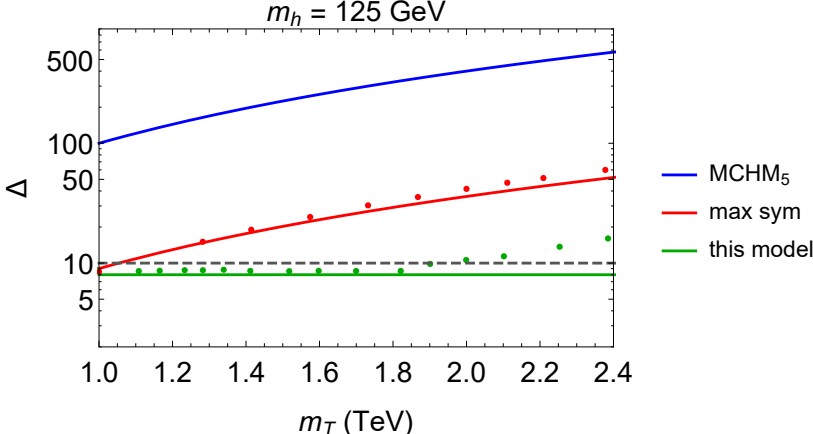

Figure 2: Comparison of the tuning $\Delta$ in several models as a function of the mass of the lightest top partner $m_T$. Solid lines correspond to the analytical estimates and dots to an actual calculation according to Eq. (46). The mass of the vector-resonance $\rho$ is assumed to obey the bound $m_\rho > 2$ TeV. The dashed gray line corresponds to $\Delta^{-1} = 10\%$.

to more tuned regions in the parameter space. Nevertheless large improvement is possible allowing to approximately double the top-partner masses at minimal tuning.

The tuning of the various models discussed above as a function of $m_T$ is presented in Fig. 2 assuming $m_\rho > 2$ TeV. Solid lines correspond to the simple analytic estimates while the dots to actual calculations using the full one-loop expression as well as the measure in (46). As we can observe, for the soft maximal symmetry case (green color) the tuning is actually flat and well approximated by (53) for $m_T \lesssim 2$ TeV, which is the maximal top partner mass that can be reached without increasing the tuning. For $m_T \gtrsim 2$ TeV, extra cancellations are required in order to correctly misalign the vacuum while keeping the Higgs light and some dependence on $m_T$ is found. Nevertheless, the model still remains rather natural: we find for instance $\Delta \simeq 15$ for $m_T \simeq 2.4$ TeV. On the other hand, by taking a more stringent cut on $m_\rho$, e.g. $m_\rho > 3$ TeV, the flat region for the tuning would shift from $\Delta \simeq 8$ to $\Delta \simeq 15$. One can clearly see how the sMCHM$_5$ with maximal symmetry allows for a minimal tuning, at the level of $\Delta^{-1} \gtrsim 10\%$, while avoiding light top partners below 2 TeV— thus escaping the current direct collider bounds. [8]

This also means that, in contrast to other minimal models, this natural CH is only just about to be tested at the HL-LHC, or later at the FCC. In fact, already before turning on the LHC, electroweak precision tests told us that $f \gtrsim 800$ GeV (see, e.g., [32] and references therein for an overview of constraints). As discussed in detail before, in the model at hand this perfectly fits with top partners above the current reach of $\sim 1.3$ TeV [17]. For the generic MCHM$_5$, on the other hand, these LHC top partner bounds already significantly cut into the parameter space that was allowed pre-LHC and push $f$ beyond a TeV, increasing the irreducible tuning. On the contrary, Fig. 2 confirms that the LHC limit on top partners does not yet drive the tuning in our model with maximally symmetric sMCHM$_5$, which is postponed to $m_T \gtrsim 2$ TeV.

Finally we would like to stress that absence of light top partners at the end of the HL-LHC program would indeed most likely be the strongest constraint on canonical composite Higgs models. The projected bounds of $m_T \gtrsim 2$ TeV [21,33] would lead to $f \gtrsim 1.6$ TeV, see Eq. (25)

---

[8]We have checked that the tuning needed for obtaining a light Higgs mass, $\max\{|\partial \log m_h^2/\partial \log x_i|\}$ with $x_i$ in Eq. (47), is always subleading with respect to $\Delta$ in Eq. (46). This tuning is the same as in the maximally symmetric MCHM$_5$ as long as $m_T \lesssim 2$ TeV, consistent with the discussion above.

(with $\epsilon = 1$), which is stronger than the projected bounds from Higgs coupling measurements, estimated to be in the ballpark of $f \gtrsim 1$ TeV [20] (see also [34] for potentially more stringent future constraints). Our model on the other hand will remain viable in the long run. Moreover, in the sMCHM$_5$ with maximal symmetry we could still expect to see effects of compositeness in Higgs couplings at the LHC, while in the conventional MCHM$_5$ this option is already disfavored by current limits from top partner searches.

# 5 Warped 5D implementation

So far we have focused on exploring the essential features of our model within the context of a 4D effective theory. While that was ideal for being able to focus on each individual aspect of the setup, the resulting model may seem somewhat *ad hoc*. In this section we present a realization of our model using a warped 5D setup, where we will see that every ingredient of the 4D model has a very natural implementation and the resulting 5D model is in fact quite simple and natural, and in no way more contrived than the original [7] holographic MCHM$_5$, but phenomenologically more successful.

The implementation of soft breaking with maximal symmetry is very simple and natural in 5D. All one needs to do is impose $SO(5)$ *universal* boundary conditions (BCs) on the bulk fields $\Psi_{l,r}$ that the SM fermions are embedded into. This means all $SO(5)$ components of the fields have the same BCs:

$$\Psi_l[+,+] = \begin{pmatrix} \chi_l \\ \bar{\psi}_l \end{pmatrix}, \Psi_r[-,-] = \begin{pmatrix} \chi_r \\ \bar{\psi}_r \end{pmatrix}, \tag{54}$$

where $\chi_l$ contains the left-handed quark doublet $q_L$ and $\psi_r$ contains $t_R$ together with the other spinors, such that one 5D bulk fermion is equivalent to a Dirac fermion containing both $\psi$ and $\chi$. As for our notation for the BCs, $\Psi_l[+,+]$ means that $\psi_l(R) = \psi_l(R') = 0$, such that $\chi_l$ contains zero modes (including $q_L$), and $\Psi_r[-,-]$ means $\chi_r(R) = \chi_r(R') = 0$. Here $R$ ($R'$) denotes the position of the UV (IR) brane.

Such universal BCs would produce a full $SO(5)$ multiplet of zero-modes for every bulk fermion, which is not viable phenomenologically. However, the superfluous modes can be lifted by introducing UV localized 2-component Weyl spinors $s_R$, $v_L$, $w_{1R}$ and $w_{2L}$ which can mix with the bulk fermions on the UV brane. To obtain the same mixing pattern as in our 4D setup the Lagrangian for the localized fields is chosen as

$$S_{\text{UV}} = \int d^4x \left\{ -i\eta_R \sigma^\mu \partial_\mu \bar{\eta}_R - i\bar{\xi}_L \bar{\sigma}^\mu \partial_\mu \xi_L + \frac{1}{\sqrt{R}} \chi_l(R) M_R^\dagger \eta_R + \frac{1}{\sqrt{R}} \psi_r(R) M_L^\dagger \xi_L + \text{h.c.} \right\}, \tag{55}$$

where the $\eta_R$, $\xi_L$ are the same fields as in (34) and $M_{R,L}$ are the the dimensionless mass matrices analogous to (36). The masses are now in fact measured in units of $R \sim 1/M_{\text{Pl}}$ and are replaced by the dimensionless quantities $\mu_i = m_i R$.

Note that in (55) we are again forbidding couplings of the type $\eta_R - \xi_L$ and $\chi_l - \psi_r$ in order to avoid reintroducing the double tuning, as discussed below (37). In the 5D version this can be enforced (similarly to Sec. 3) by introducing a $Z_2$ symmetry under which the entire bulk Dirac $\Psi_l$ multiplet (both $\chi_l$ and $\psi_l$) as well as $\eta_R$ have negative parity, while the other fields have positive parity. This $Z_2$ symmetry will only be broken on the IR brane, where the $\chi_l - \psi_r$ terms are necessary to give mass to the SM quarks. In fact the starting boundary conditions on the IR brane in (54) will be modified by the presence of the following IR–localized action,

$$S_{\text{IR}} = -\int d^4x \left( \frac{R}{R'} \right)^4 \left( \lambda \, \chi_l(R') V \psi_r(R') + \text{h.c.} \right), \tag{56}$$

where $V$ is the Higgs parity operator, see below (35), and $\lambda$ is an $\mathcal{O}(1)$ free parameter. The form of (56) is consistent with maximal symmetry [19].

The UV action in (55) corresponds to the first two terms in (35) (plus kinetic terms) and the explicit breaking of the Higgs shift symmetry is fully encoded in the dimensionless matrices $M_{R,L}$. Brane localized fields analogous to $\eta_R$ and $\xi_L$ were actually already considered in [35] as classical Lagrangian multipliers to enforce the desired BCs in the holographic approach: the soft breaking setup can thus be seen as making those fields dynamical and controlling their impact through their masses $\mu_i$. In the limit of large $\mu_i$, $\eta_R$ and $\xi_L$ are in fact true Lagrange multipliers, enforcing opposite BCs for the $SO(5)$ components not corresponding to SM fermions. In this limit, all results from conventional holographic composite Higgs models are recovered. However, one can now interpolate between true zero modes for the new vector-like quarks ($\mu_i = 0$) and pure Kaluza-Klein (KK) excitations ($\mu_i \gg 1$) by changing $\mu_i$. For intermediate values, partially elementary KK states appear in the low energy spectrum and the model is expected to modify the Higgs potential similarly to its 4D dual.

How large values of $\mu_i$ should we choose to get a realistic model reproducing the success of the 4D picture? The most naive answer would be that $\mu_i \sim \text{TeV}/M_{\text{Pl}}$ and hence unnaturally small. However it is well-known that in 5D an effective TeV state can arise from a Planckian mass due to wave-function suppression, or, equivalently, renormalization-group running in presence of large anomalous dimension for the corresponding operator [36, 37].

To see this explicitly, let us focus on the singlet $s$, whose dimensionless mass $\mu_s$ is taken to be $\mu_s \lesssim 1$. We then consider the $SO(4)$ singlet component of $\Psi_l$, consisting of two Weyl spinors, $s_L \in \chi_l$ and $\sigma_R \in \psi_l$, that are KK-decomposed as:

$$s_L(x, y) = \sum_n g_n(y)\chi_n(x), \quad \bar{\sigma}_R(x, y) = \sum_n f_n(y)\bar{\psi}_n(x), \tag{57}$$

where $\chi_n(x)$ and $\psi_n(x)$ solve the 4D Dirac equation with mass $m_n$ and $g_n(y), f_n(y)$ are the bulk profiles. One also needs to expand the brane-localized field, $s_R \in \eta_R$, in the same basis, in order to account for the mixing in (55) with the 5D field:

$$\bar{s}_R(x) = \sum_n e_n \bar{\psi}_n(x). \tag{58}$$

The presence of the UV action (55) modifies the BCs for the bulk fields as

$$f_n(R) = 0 \to f_n(R) - \frac{\mu_s^2}{m_n R}g_n(R) = 0, \tag{59}$$

whereas $f_n(R') = 0$ is unaffected (we are for now neglecting all effects from the IR brane). One can derive an approximate formula for the mass of the lightest resonance, $m_1$, in the limit $m_1 R' \lesssim 1$ (see e.g. [38])

$$m_1^2 \sim \begin{cases} (2c_l - 1)\mu_s^2 R^{-2} & c_l > 1/2 \Rightarrow \text{UV} \\ \frac{(1-4c_l^2)\mu_s^2}{|1+2c_l-\mu_s^2|}R'^{-2}\left(\frac{R}{R'}\right)^{-1-2c_l} & c_l < 1/2 \Rightarrow \text{IR} \end{cases}, \tag{60}$$

where $c_l$ is the (5D) bulk mass of $\Psi_l$ in units of $1/R$. In case of UV localized zero modes (corresponding to $c_l > 1/2$), the mass of this state is indeed given by the mass term on the UV brane. Unless $\mu_s$ is tuned to be tiny, $\mu_s \sim 10^{-16}$, the UV-localized spinor $s_R$ decouples from the low energy theory and this model would be indistinguishable from the conventional holographic Higgs.

However a TeV scale state with a sizeable overlap with the elementary spinor $s_R$ can naturally emerge in case of deep IR localization, namely $2c_l + 1 \approx 0$, corresponding to a (partially-)

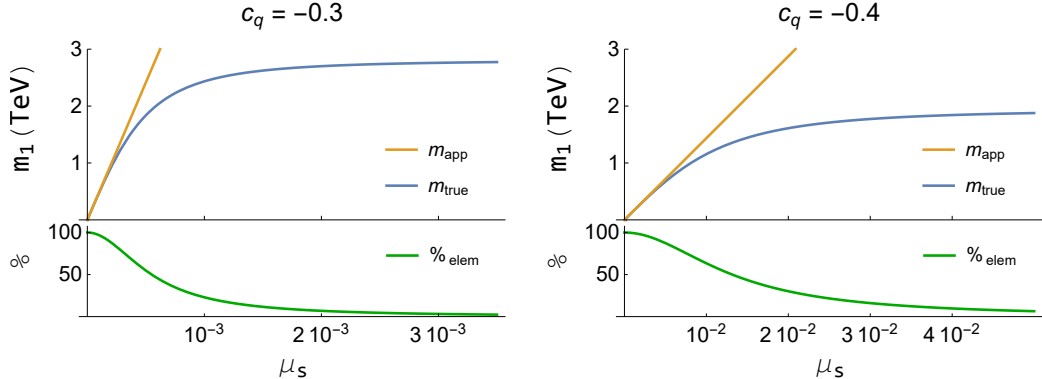

Figure 3: The mass of the lightest state, $m_1$, as a function of $\mu_s$ for $c_l = -0.3$ (left) and $c_l = -0.4$ (right). The blue line is the true numerical value, that is compared with the approximate formula (60) in orange. As expected, the two lines depart at $m_1 R' \approx 1$. The green line represents the overlap of this state with the UV-localized spinor $s_R$. When such overlap becomes negligible, the $[-, +]$ BC is effectively recovered and there is no elementary state in the spectrum at low energy.

elementary lightest KK mode, allowing to lift the full spectrum. Notice that, since the occurrence of such a partially-elementary state is linked to the presence of IR-localized zero modes, it is relevant only for third generation fermions, which are exactly those that usually come with light partners. Hence the issue of light partners is getting naturally resolved in this setup, no additional tweaking of the model is needed.

The comparison between the approximate formula (60) shown in orange and the true numerical result shown in blue is displayed in Fig. 3 for $c_l = -0.3$ (left) and $c_l = -0.4$ (right), where $R' = 1/3 \, \text{TeV}^{-1}$. We see a good agreement up to $m_1 R' \sim 1$, as expected. The green lines in Fig. 3 show the elementary content of the lightest state with mass $m_1$, corresponding to the numerical value of $e_1^2$ in (58) after canonically normalizing $\psi_1(x)$. As expected it is almost completely elementary in the $\mu_s \to 0$ limit and becomes mostly composite for large values of $\mu_s$ which approach the limit of the $[-, +]$ BC. We notice that the closer $c_l$ is to $-0.5$, the more natural the value of $\mu_s$ can be: for $c_l = -0.4$ a largely elementary state with $\sim 1 \, \text{TeV}$ mass is realized for $\mu_s \sim 0.01$, whereas $c_l = -0.3$ requires $\mu_s \sim 0.001$, but allows for $m_1 \gtrsim 2 \, \text{TeV}$. In general, we see that for larger values of $\mu_s$, the elementary state "migrates" towards higher KK excitations and decouples. Slightly smaller values of $\mu_s$ show instead a sizeable elementary component for the lightest excitation, thus realizing the 4D low-energy theory discussed in the previous section.

Notice that due to the almost complete IR localization, the mass of this state is very similar to the mass of a light custodian from a $[-, +]$ BC to which it asymptotes, $m_1 \lesssim m_{\text{cust}}$ [8]. Of course, even though the mass is similar, this state is substantially different from a light custodian due to its degree of 'elementariness' and its correspondingly different impact on the Higgs potential. If one wants to keep $\mu_s \in (0.001, 0.01)$ and therefore $c_l \in (-0.4, -0.3)$, we need to raise $R'$ to compensate the suppression typical of a light custodian, if we want to realize $m_1 \gtrsim 2 \, \text{TeV}$. This is the reason behind the choice of $R' = 1/3 \, \text{TeV}^{-1}$ in Fig. 3. With $f = 800 \, \text{GeV}$, such value of $R'$ implies $g_* \sim 7.5$ and thus $N_{\text{CFT}} \sim 3$.[9] Moreover, we have checked that the top mass can be successfully reproduced in the presence of partially elementary KK states, confirming the findings in Ref. [17] for the 4D model.

From the 5D perspective, the raising of the top partner masses is achieved by the softened global symmetry breaking allowing for less extreme IR localization of the top for a fixed $R'$

---

[9]Of course, allowing for (technically natural) smaller $c_s$ makes possible to keep $R' = 1 \text{TeV}^{-1}$.

and top and Higgs masses (or if one fixes the localization then $R'$ can be raised). This opens the possibility to go beyond the small $m_{\text{cust}}$ of the minimal MCHM$_5$. Finally, let us mention that these results hold in the very minimal setup, where no localized brane kinetic terms for bulk fields are included and the effects of the IR-brane localized terms are neglected. These additional ingredients are expected to make the model more flexible and able to reproduce all details of the 4D scenario beyond the general agreement shown here. Such a study, including the calculation of the Higgs potential and a detailed survey of the phenomenology in the dual 5D scenario is left for future work [39].

# 6 Conclusion

While the composite Higgs scenario is one of the most attractive ideas to solve the hierarchy problem, non-discovery of the top and gauge partners at the LHC is forcing the traditional incarnations into ever more tuned regions. In this paper we presented a very simple modification of the minimal model which is naturally evading all LHC bounds and is able to get away with tuning at the $\lesssim 10\%$ level. The main ingredient is to use complete multiplets under the global symmetry both for the elementary and the composite states. In practice this means combining the soft breaking approach with that of maximal symmetry, which turns out to be a perfect match. Maximal symmetry removes the double tuning while soft breaking raises the top partners, allowing a complete natural model to emerge. The utility of these ideas becomes most clear in the 5D picture, where it actually corresponds to a very simple modification of the boundary conditions used in the minimal model. Choosing all bulk fermions to have universal UV and IR boundary conditions (along with some localized UV brane degrees of freedom) will automatically lead to the successful 4D picture laid out earlier. The resulting simple model is perfectly consistent with a low $f = 800\,\text{GeV}$ and a natural spectrum of heavy resonances and puts us back to the level of tuning of the LEP era.

# Acknowledgments

We are grateful to Ofri Telem for collaborations at the early stages of this project, to Julian Bollig for useful discussions on the 5D analysis, and to Kaustubh Agashe for comments and discussions. S.B. thanks the particle theory group at Cornell for its hospitality while this work was initiated. The research of C.C. was supported in part by the NSF grant PHY-1719877 and also supported in part by the BSF grant 2016153.

# A Trigonometric parity and soft breaking

In this Appendix, we discuss the fate of trigonometric parity in the soft-breaking setup of (35). For this, recall that the trigonometric parity $\sin(h/f) \longleftrightarrow -\cos(h/f)$ can be defined as the following discrete symmetry [18]:

$$\Sigma \to V\Sigma P', \quad \Psi_L \to P\Psi_L, \quad \Psi_R \to VPV\Psi_R, \quad \psi_L \to V\psi_L, \quad \psi_R \to P'\psi_R, \qquad (61)$$

where $P = \text{diag}(1,1,1,\sigma_1)$, $P' = \text{diag}(1,1,1,-\sigma_3)$.

The transformation (61) would be a symmetry of the Lagrangian (35) if $M_{L,R}$ were to transform as

$$M_R \to VM_R, \quad M_L \to P'M_L \Rightarrow \Gamma_R \to V\Gamma_R V, \quad \Gamma_L \to P'\Gamma_L P', \qquad (62)$$

where $\Gamma_{L,R}$ are defined in (38).

Whether trigonometric parity is eventually preserved or not depends on the spurion vacuum expectation values (namely, on their explicit form in (38)). The parity-conserving vacuum is found by solving

$$\Gamma_R = V\,\Gamma_R\,V \quad \text{and} \quad \Gamma_L = P'\,\Gamma_L\,P'. \tag{63}$$

The first condition is always trivially satisfied, while the second condition implies

$$(m_v - m_{w_2})(m_v + m_{w_2}) = 0. \tag{64}$$

Moreover, we notice that there is another way to implement trigonometric parity in addition to (61), namely interchanging the left and right chiralities in (61):

$$\Sigma \to V\,\Sigma\,P', \quad \Psi_L \to VPV\,\Psi_L, \quad \Psi_R \to P\Psi_R, \quad \psi_L \to P'\psi_L, \quad \psi_R \to V\,\psi_R. \tag{65}$$

Similar arguments then imply the following spurion transformations:

$$\Gamma_R \to P'\,\Gamma_R\,P', \quad \Gamma_L \to V\,\Gamma_L\,V, \tag{66}$$

so that another parity-preserving vacuum exists if

$$\Gamma_R = P'\,\Gamma_R\,P' \quad \text{and} \quad \Gamma_L = V\,\Gamma_L\,V. \tag{67}$$

The second condition is satisfied identically, while the first one requires:

$$m_{w_1}^2 = 0. \tag{68}$$

Combining (64) and (68), we conclude that trigonometric parity is a true symmetry of the theory if and only if

$$\boxed{(m_v^2 - m_{w_2}^2)m_{w_1}^2 = 0.} \tag{69}$$

Thus, we can see that $\xi = 0.5$ can be avoided in our setup within the fermion sector for generic values of the fermion masses. Moreover, this way of breaking trigonometric parity still ensures that double tuning is avoided, see the discussion below (40).

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
