# Peer review of "A Natural Composite Higgs via Universal Boundary Conditions"

_SciPost Physics, doi:SciPost Phys. 10, 121 (2021)_

## Round 2 · Referee Report · Anonymous (Referee 1) · 2020-9-3

Report

The paper proposes a new holographic (5d) realisation of the Composite Higgs scenario. In the new model the amount cancellation (fine-tuning) required to keep the EWSB scale "v" separated from the Higgs Goldstone decay constant "f" is at the level of (f/v)^2, which is believed to be the minimal possible tuning in these constructions. At the same time, and this is what makes the model different from other departures from the "Minimal" model, the model does not require extra tuning to avoid unobserved light Top Partner fermion resonances for the Higgs being as light as we observe it.

I have two concerns about the relevance of the paper. The first one is phenomenological as it is unclear if anomalously light top partner direct searches will be or not the strongest probes of Higgs compositeness at the end of the HL-LHC. Departures of the Higgs couplings from the SM predictions, of order (v/f)^2 and hence directly related with the "minimal" tuning, will be probed accurately. I have not found mention of this aspect in the manuscript, but it seems clear that this kind of effects are not reduced in the model under consideration relative to the generic expectation.

The second concern is that the phenomenological virtues of the model do not come from robust structural assumptions or emergent phenomena that are deeply rooted in the theory. The first ingredient of the model is Maximal Symmetry, reviewed in section 2. This is an SO(5)' group, different from the SO(5) Goldstone symmetry, that acts on the fermionic composite resonances and that is enforced by a choice of the composite resonance masses. I believe that this symmetry can not be promoted to a true symmetry of the Composite Sector of the theory. Or, equivalently, the choice of the mass spectrum that respects the symmetry cannot be viewed as the consequence of some global symmetry that is respected by the underlying Composite Sector Dynamics. I think it is pretty obvious that the symmetry cannot be uplifted to a true symmetry of the theory, and the manuscript does not claim otherwise.

However the manuscript claims that the symmetry can be extended also to the composite fermions-Higgs interaction, by acting on the Goldstone Boson Higgs as in eq.(2.13). I am confused by that equation. The Sigma matrix is not a generic matrix, but it is of the form (2.9) and the only degrees of freedom it contains are the Goldstone scalars that are present in the U Goldstone matrix. I do not see how the transformation in eq.(2.13) could be realised as a transformation acting only on those degrees of freedom. Actually it seems to me that eq.(2.13), using eq.(2.11) for V, can be turned into the operation U^2--> gL U^2 gL^\dagger, with gL a generic SO(5) matrix, which is clearly not permitted since U only contains the exponent of broken generators. This aspect should be definitely clarified if the manuscript has to be further considered for publication.

The second element of the model is Goldstone symmetry "soft breaking", also reviewed in Section 2. This is based on adding extra elementary degrees of freedom that fill a five-plet of the SO(5) Goldstone symmetry and couple them to the Composite Sector in a way that respects the symmetry. The SM fermions Yukawa coupling and the explicit symmetry breaking emerge from elementary-to-elementary mass-mixings. The setup allows to relax the connection between the Higgs mass and the mass of the Top Partners. It is technically correct to add extra degrees of freedom to the theory, and it is technically correct to assign to them transformation properties under SO(5) and postulate a specific patter of breaking of the symmetry. However it is unclear which microscopic assumption on the underlying theory would produce the advocated symmetry structure: the elementary fermions know nothing about the SO(5) global symmetry of the composite constituents, a priori.

The structural considerations above are not made explicit in the manuscript. It is admitted that the required structure is somehow ad hoc, but this is not sufficient. Being or not "ad hoc" can be matter of taste. Being or not able to sharply state the microscopic UV theory assumptions that produce the desired effect in the low-energy theory is an objective statement. Emphasis is given in the manuscript to the "simplicity" of the 5d holographic model that realises all these required structures. On one hand I agree that from the 5d perspective the model is not more involved than the ordinary Minimal Composite Higgs Model (MCHM). On the other hand the MCHM can be viewed as a generic realisation of a clear set of well-defined microscopic assumptions on the Composite and Elementary sectors of the 4d underlying theory. The new ingredients of the present model instead (Maximal Symmetry, in the first place) do not emerge from microscopic assumptions.

In conclusion, I am not enthusiastic about the paper. In itself, the fact that minor deformations and/or special choices of the MCHM parameters can reduce the fine tuning and lift the Top Partners, is not surprising. It would have been interesting if these choices/deformations had been found to emerge from structural microscopic assumptions, which is not the case here. On the other hand it is interesting to see one concrete example, and definitely a lot of work is done in the paper to explain why the non-generic features introduced in the construction (in the first place, the fact that the IR boundary conditions do not split SO(4) multiplets, in order to produce Maximal Symmetry) do indeed produce an interesting phenomenology.

So I believe that the paper can be published, provided the limitations are more clearly spelled out.

Requested changes

1-One should discuss the impact of Higgs coupling measurements and state if or not the model behaves differently from the standard one.

2-The meaning of Eq.(2.13) should be clarified, in light of the comment made in the report.

3-It should be stated clearly which aspects of the construction are structural assumption about the underlying microscopic theory and which ones are not. in light of the remarks made in the report.

  • validity: high
  • significance: good
  • originality: good
  • clarity: ok
  • formatting: -
  • grammar: -

Author:  Simone Blasi  on 2020-12-29  [id 1117]

(in reply to Report 1 on 2020-09-03)
Category:
remark
answer to question
correction
validation or rederivation
pointer to related literature

We thank the referee for the detailed report that helped us improving our manuscript. We appreciate the very careful reading and the time spent on understanding the details of our proposal. In the following, we discuss the points raised in the report in detail and provide our answers together with the corresponding changes in the text.

The first concern of the referee is on whether anomalously light top partners will be the most stringent probe for composite Higgs at the end of HL-LHC. Indeed the projections for the long-term run predict bounds on mT of around 2 TeV, see e.g. arXiv:1810.08954 and arXiv:1409.0100, which according to Eq.(2.25) (with eps=1) corresponds to f >1.6 TeV. In contrast, Higgs coupling extractions are expected to set bounds around f > 1 TeV, see the horizontal HL-LHC line in the left plot of Fig 3.3 of arXiv:1502.01701. This also agrees well with the generic Delta = v^2/f^2 = 5% estimate for the precision of Higgs coupling extractions. In consequence, bounds on light top partners will be a major driving factor for limits on composite Higgs scenarios. To stress this, and address the referee's question, we added a paragraph on p.17, right before Section 5.

The referee also raised a concern regarding the relation between maximal symmetry and the Goldstone matrix Sigma. We believe that this concern appeared due to our misleading formulation in the text, which we have now rectified around Eq.(2.15). The transformation in Eq.(2.13), Eq.(2.15) in the new version, was merely meant to be a spurious transformation of the Sigma matrix (which for this purpose is treated as a generic 5x5 matrix) that we introduce in order to have an easier way of performing the spurion analysis. However, it was never our intention to claim that maximal symmetry can be realized as a true transformation acting on the Goldstone degrees of freedom. We further clarified the difference between a "proper" SO(5) transformation of the Sigma matrix, which can indeed be seen as a transformation acting on the Goldstone dofs, and a "spurious" SO(5)' transformation in the caption of Table 1. We hope that these changes will satisfy the referee's concern on this point.

Another ongoing concern of the referee throughout the report is whether the ingredients employed in our model could arise from assumptions on the microscopic theory. Regarding the hypothesis of soft breaking, we do believe that it can be formulated in terms of a true symmetry of the microscopic theory. The assumption one needs to make is that interactions between the elementary fields and the fundamental constituents of the strong sector respect a global symmetry, which is broken only via soft mass terms in the elementary sector. We believe that this statement does correspond to a clear microscopic assumption, and we would also like to stress that the exotic degrees of freedom that we introduce in the 4d effective theory are not generic vector-like fermions that are chosen arbitraily, but exactly those required in order to comply with the assumption of SO(5)-symmetric partial compositeness. On the other hand, a concrete UV completion of composite Higgs models with partial compositeness in 4d is notoriously challenging, especially for the minimal SO(5)/SO(4) coset. Despite this well-known difficulty, we see nothing preventing the microscopic interpretation above from being conceptually applicable to composite Higgs models. We added a paragraph on this at page 8 in the manuscript.

The referee also states that the elementary fermions know nothing, a priori, about the SO(5) symmetry. Although we agree with this statement, it is clear that the elementary fermions do know about the SO(5) symmetry in the best realization of the SO(5)/SO(4) setup, namely the extra-dimensional picture, as all the degrees of freedom genuinely come as full SO(5) representations due to the bulk gauge symmetry. Following this line of reasoning, it is straightforward to reintroduce the missing degrees of freedom in the elementary side, that would indeed correspond to SO(5)-universal boundary conditions as discussed below Eq.(5.1), and examine their interplay with the composite resonances as done in Section 3 and 4. On the other hand, the holographic picture further supports the fact that our construction does have a microscopic interpretation: the 5d model in Section 5 may be interpreted as a UV completion on its own (up to a String Theory completion) or even in the 4d sense due to the AdS/CFT correspondence.

The other ingredient that we employ extensively is maximal symmetry. While we agree again with the referee that this symmetry is not present in the microscopic description of the theory, we would like to emphasize that it is nothing but an emergent accidental symmetry of the composite sector. Whether or not such a symmetry actually appears will depend on the details of the dynamics of the theory, which condensates actually form, etc. It is true that it can not be enforced by imposing a specific symmetry on the microscopic UV description, but it does not make it less of a symmetry in case it does emerge. This can be nicely translated again into the extra dimensional picture - the boundary terms for the fermions on the IR brane do not have to obey the SO(5)' symmetry, but they surely can (for clarification we have added the IR-localized action for our model in the text in Eq.5.3). In that case there will be an actual enhanced global symmetry of the composite sector, interpreted as an accidental IR symmetry. Such symmetries do happen in cases where we can reliably follow the dynamics of strongly coupled gauge theories (for example in supersymmetric gauge theories). But of course we agree with the basic statement that it is not guaranteed by the UV structure of the theory. We added a corresponding statement in a footnote on page 7.

We hope that we have satisfactorily addressed all the concerns of the referee with our changes and our explanations here, and that our paper improved by the requested clarifications, will now be suitable for publication.

---

## Round 3 · Referee Report · Anonymous (Referee 1) · 2021-2-3

Report

The clarifications added in the current version do indeed improve the quality of the manuscript. I have some followup comment, reported below, to the authors' reply. Based on these comments the authors might want to make some further adjustment to the text. But I leave the choice to them.

Concerning Higgs coupling constraints: the most up-to-date HL-LHC Higgs coupling measurements projections are reported in https://arxiv.org/pdf/1905.03764.pdf. In particular one can look at Table 9 (in particular, to the single-operator sensitivity to $c_\phi$, which is the relevant one in composite Higgs at large $g_*$). This is quite stronger than what previously considered to be possible, and correspondingly the reach on $f$ one extracts from Figure 7 left panel is higher than the $1.6\,$TeV mentioned in the manuscript for the reach from top partners direct searches . In the text describing the table in https://arxiv.org/pdf/1905.03764.pdf it is explained that the sensitivity to $c_\phi$ reported in the table is still subject to large uncertainties. However I guess this suggests that one should be carful before stating with certitude that top partners direct searches will be the strongest probe of Composite Higgs at the end of the HL-LHC.

Concerning the microscopic origin of the soft-breaking structure: it is indisputable that elementary degrees of freedom having to transform under the global group of the composite sector restricts the viable options for the microscopic origin of the symmetry. In particular, the symmetry cannot emerge anymore as an Accidental (in the proper sense detailed below) Flavour-type symmetry of the Composite Sector.

Finally, about Maximal symmetry being possibly "Accidental": I really do not understand what this means. There exist a sharp notion of Accidental Symmetries, based on operator classification at a given order (like baryon and lepton number in the SM). Maximal Symmetry definitely does not emerge in this way. I am not sure that the authors have something concrete in mind when they refer to the dynamics of the composite sector, to which condensates form or do not form. In any case, I recommend them not to use the word "accidental" to describe what they have in mind because this is very confusing.

On general grounds, it is technically legitimate to state that the Holographic model is a "UV completion". However it is so in a rather limited sense, due to the limited cutoff of the Holographic model and to the lack of a String origin of the Holographic model itself. The ingredients employed in the model do indeed make sense in the Holographic setup. While they make less sense in the perspective of a possible UV-completion based on strong dynamics in 4d and extending up to energies well above the TeV scale. This is in my view the main limitation of the work, which however remains interesting especially with the clarifications included in the revised version.
  • validity: -
  • significance: -
  • originality: -
  • clarity: -
  • formatting: -
  • grammar: -

Author:  Simone Blasi  on 2021-05-04  [id 1406]

(in reply to Report 1 on 2021-02-03)
Category:
remark
answer to question

We are glad that our clarifications have met the points raised by the Referee. We here briefly answer the follow-up comments.

We acknowledge the recent improvements in the possible extraction of the compositeness scale f at the HL-LHC and we have added a reference to it in the text.

We agree that the nature of the global symmetry in our model may differ from conventional Composite Higgs models, as soft breaking requires a deeper connection between the elementary fermions and those charged under the confining group. The question of whether the global symmetry will be accidental or not in such UV theory is beyond the scope of our investigation. We however remark that such a connection is not implausible or particularly far-fetched: in the Standard Model something very similar happens since quarks and leptons share global (and local) symmetries, but just quarks are charged under the condensing strong force.

Finally, it is well-known that theories can have an emergent global symmetry in the IR which was not realized in the UV. The simplest example is a coupling that is O(1) at high scales and runs to zero at low energies, implying a larger global symmetry which we may refer to as accidental. However, in order to avoid confusion with the same terminology in the context of EFTs, we have accordingly changed the wording in the footnote 4.

---

## Round 3 · List of Changes

1. Discussion on the expected bounds from HL-LHC on composite Higgs models in terms of light top partners vs Higgs couplings (page 17).

  2. Clarification of the spurious nature of the SO(5)' transformation acting on the Goldstone matrix (page 6 and caption of Table 1).

  3. Paragraph added on the possible microscopic interpretation of soft breaking (page 8).

  4. Clarification of the assumptions concerning maximal symmetry (footnote at page 7).

  5. Inclusion of the explicit form of the IR-localized action considered in Section 5 (Eq. 5.3 at page 19).

---

## Round 4 · Referee Report · Anonymous (Referee 3) · 2021-5-12

Report

As in the previous report, I confirm that the paper is suited for publication. I thank the authors for further clarifying some aspects related with my follow-up comments.

---

## Round 4 · List of Changes

1. Reference added on the prospects for the extraction of the compositeness scale at the HL-LHC (Ref. [34])

2. Wording changed in the footnote 4 regarding maximal symmetry

---

## Editorial Decision

published